# Neural encoding with unsupervised spiking convolutional neural network

Chong Wang[1,2,3], Hongmei Yan [2,3✉], Wei Huang[2,3], Wei Sheng[2,3], Yuting Wang[2,3], Yun-Shuang Fan[2,3], Tao Liu[2], Ting Zou[2], Rong Li [2,3✉] & Huafu Chen [1,2,3✉]

Accurately predicting the brain responses to various stimuli poses a significant challenge in neuroscience. Despite recent breakthroughs in neural encoding using convolutional neural networks (CNNs) in fMRI studies, there remain critical gaps between the computational rules of traditional artificial neurons and real biological neurons. To address this issue, a spiking CNN (SCNN)-based framework is presented in this study to achieve neural encoding in a more biologically plausible manner. The framework utilizes unsupervised SCNN to extract visual features of image stimuli and employs a receptive field-based regression algorithm to predict fMRI responses from the SCNN features. Experimental results on handwritten characters, handwritten digits and natural images demonstrate that the proposed approach can achieve remarkably good encoding performance and can be utilized for "brain reading" tasks such as image reconstruction and identification. This work suggests that SNN can serve as a promising tool for neural encoding.

[1] The Center of Psychosomatic Medicine, Sichuan Provincial Center for Mental Health, Sichuan Provincial People's Hospital, University of Electronic Science and Technology of China, Chengdu 611731, China. [2] School of Life Science and Technology, University of Electronic Science and Technology of China, Chengdu 610054, China. [3] MOE Key Lab for Neuroinformation; High-Field Magnetic Resonance Brain Imaging Key Laboratory of Sichuan Province, University of Electronic Science and Technology of China, Chengdu 610054, China. ✉email: hmyan@uestc.edu.cn; rongli1120@gmail.com; chenhf@uestc.edu.cn

The objective of neural encoding is to predict the brain's response to external stimuli, providing an effective means to explore the brain's mechanism for processing sensory information and serving as the foundation for brain–computer interface (BCI) systems. Visual perception, being one of the primary ways in which we receive external information, has been a major focus of neural encoding research. With the advancement of non-invasive brain imaging techniques, such as functional magnetic resonance imaging (fMRI), scientists have made remarkable progress in vision-based neural encoding[1–4] over the past two decades, making it a hot topic in neuroscience.

The process of vision-based encoding typically involves two main steps: feature extraction and response prediction[5]. Feature extraction aims to produce visual features of the stimuli by stimulating the visual cortex. An accurate feature extractor that approximates real visual mechanisms is crucial for successful encoding. Response prediction aims to predict voxel-wise fMRI responses based on the extracted visual features. Linear regression[6] is commonly used for this step, as the relationship between the features and responses should be as simple as possible. Previous studies have shown that the early visual cortex processes information in a manner similar to Gabor wavelets[7–9]. Building on this finding, Gabor filter-based encoding models have been proposed and successfully applied in tasks such as image identification and movie reconstruction[1,3]. In recent years, convolutional neural networks (CNNs) have garnered significant attention due to their impressive accomplishments in the field of computer vision. Several studies[10,11] have utilized representational similarity analysis[12] to compare the dissimilarity patterns of CNN and fMRI representations, revealing that the human visual cortex shares similar hierarchical representations to CNNs. As a result, CNN-based encoding models have become widely used and have demonstrated excellent performance[2,4,13,14]. However, it is important to note that despite the success of CNNs in encoding applications, the differences between CNNs and the brain in processing visual information cannot be overlooked[15].

In terms of computational mechanisms, a fundamental distinction exists between the artificial neurons in CNNs and the biological neurons, whereby the former propagate continuous digital values, while the latter propagate action potentials (spikes). The introduction of spiking neural networks (SNNs), considered the third generation of neural networks[16], has significantly reduced this difference. Unlike traditional artificial neural networks (ANNs), SNNs transmit information through spike timing. In SNNs, each neuron integrates spikes from the previous layer and emits spikes to the next layer when its internal voltage surpasses the threshold. The spike-timing-dependent plasticity (STDP)[17,18] algorithm, which is an unsupervised method for weight update and has been discovered in mammalian visual cortex[19–21], is the most commonly used learning algorithm for SNNs. Recent studies have applied STDP-based SNNs to object recognition and achieved considerable performance[22–24]. The biological plausibility of SNNs provides them with an advantage in neural encoding.

In this paper, a spiking convolutional neural network (SCNN)-based encoding framework was proposed to bridge the gap between CNNs and the realistic visual system. The encoding procedure comprised three steps. Firstly, a SCNN was trained using the STDP algorithm to extract the visual features of the images. Secondly, the coordinates of each voxel's receptive field in the SNN feature maps were annotated based on the retinal topological properties of the visual cortex, where each voxel receives visual input from only one fixed location of the feature map. Thirdly, linear regression models were built for each voxel to predict their responses from corresponding SNN features. The framework was evaluated using four publicly available image-fMRI datasets, including handwritten character[25], handwritten digit[26], grayscale natural image[1], and colorful natural image datasets[27]. Additionally, two downstream decoding tasks, namely image reconstruction and image identification, were performed based on the encoding models. The encoding and decoding performance of the proposed method was compared with that of previous methods.

## Results

**Encoding performance on handwritten character dataset**. We built SCNN-based encoding models (see Fig. 1a) on four image-fMRI datasets and realized image reconstruction and image identification tasks based on the pre-trained encoding models (see Fig. 1b, c). Table 1 provides the basic information about these datasets, and details can be found in Methods. To predict the fMRI responses evoked by handwritten characters, the SCNN was first constructed using the images in the TICH dataset (with the exclusion of images in the test set and the inclusion of 14,854 images for the 6 characters). This was done to maximize the representation ability of the SCNN. Subsequently, voxel-wise linear regression models were trained with the fMRI data in the train set for each participant. The encoding performance was measured using Pearson's correlation coefficients (PCC) between the predicted and measured responses to the test set images. Moreover, the proposed model was compared with a CNN-based encoding model, where the network architecture of CNN was constrained to be consistent with that of the SCNN (Supplementary Table 1). The CNN was trained using the Adam optimizer[28] with a learning rate of 0.0001 for 50 epochs on the TICH dataset, achieving a classification accuracy of 99% on the test set images. The subsequent encoding procedures for CNN were identical to those for SCNN. To eliminate the influence of noise (unrelated to the visual task) voxels on the result, 500 voxels with the highest encoding performance for each subject were selected for comparison. Figure 2a displays the prediction accuracies for SCNN and CNN-based encoding models. The results indicate that the accuracies of SCNN on all three subjects were significantly higher than those of CNN ($p < 10^{-18}$, one-tailed, two-sample $t$-test). This finding suggests that the SCNN has greater potential than CNN for encoding tasks.

The degree of involvement of a voxel in the visual task is a determining factor in its predictability. Specifically, if a voxel receives a substantial amount of stimulus information, its fMRI activities will be more predictable, and vice versa. To validate this hypothesis, we visualized the distributions of stimulus intensities and voxel receptive fields. By annotating the receptive field for each voxel through threefold cross-validation on the train set data, the top 100 voxels with the highest $R^2$ of each participant were selected for analysis. The mean stimulus intensities of the train set and the receptive fields of the selected voxels are shown in Fig. 2b, c. Their spatial distribution patterns, which approximately followed Gaussian distributions along the $x$-axis and uniform distributions along the $y$-axis, were found to be quite similar. This suggests that the receptive fields of these informative voxels tended to be located in areas with higher stimulus intensity. This finding provides further evidence of the efficacy of the receptive field-based feature selection algorithm employed in this study.

**Encoding performance on handwritten digit dataset**. To verify the encoding performance of the proposed approach on handwritten digit stimuli, we trained the SCNN using 2000 prior images that were not utilized in the fMRI experiment. Voxel-wise encoding models were then constructed on the train set of this dataset. Similarly, CNN-based encoding models were built on the

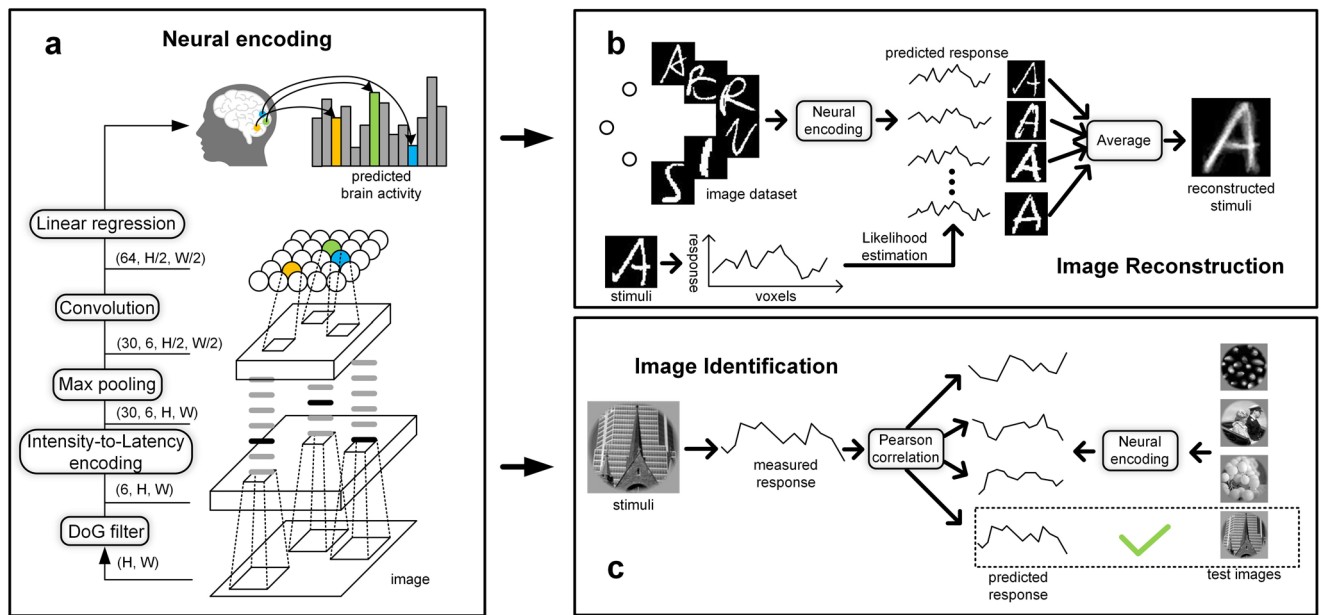

**Fig. 1 The flow chart of the encoding and decoding processes. a** The illustration of the encoding model. The proposed model uses a two-layer SCNN to extract visual features of the input images and uses linear regression models to predict the fMRI responses for each voxel. **b** The diagram for the image reconstruction task, which aims to reconstruct the perceived images from the brain activity. The handwritten character images are adapted from the TICH character dataset[47] with permission. **c** The diagram for the image identification task, which aims to identify which image is perceived based on the fMRI responses. The grayscale natural images are reproduced with permission from Kay et al.[1].

**Table 1 The basic information of the image-fMRI datasets.**

| Dataset | Subjects | ROI | Samples (train/test) |
|---|---|---|---|
| Handwritten Character | 3 | V1, V2 | 270/90 |
| Handwritten Digit | 1 | V1, V2, V3 | 90/10 |
| Grayscale Natural Image | 2 | V1, V2, V3 | 1750/120 |
| Colorful Natural Image | 5 | V1, V2, V3 | 1200/50 |

handwritten digit dataset, and the top 500 voxels with the highest encoding performance were selected for comparison. The encoding results are presented in Fig. 2d, and the results indicate that the encoding accuracies of SCNN were significantly higher than those of CNN ($p = 6.78 \times 10^{-18}$, one-tailed two-sample $t$-test).

**Encoding performance on natural image datasets**. In comparison to handwritten characters and digit images, natural images are more intricate and closely resemble our everyday visual experiences. To assess the feasibility of the proposed approach for encoding natural image stimuli, we trained and tested the encoding model on grayscale and colorful natural image datasets. The SCNNs utilized for encoding were trained on the train set images of these datasets.

For the grayscale natural image dataset, the utilization of task-optimized CNN-based encoding models is not feasible due to the absence of category labels in the visual stimuli. A comparison was conducted between our approach and the Gabor wavelet pyramid (GWP) model proposed by Kay et al.[1], as well as the brain-optimized CNN (GNet)[13,29]. Instead of classifying the input images, the CNN in GNet was trained to predict the fMRI responses in an end-to-end fashion. The architecture of GNet can be found in Supplementary Table 2. Independently, we trained the GNet for each visual area in each subject (a total of 6 models were trained). Regions of interest (ROI)-level analysis was performed on this dataset, and for each visual area, 200 voxels

with the highest encoding performance (100 for each subject) were selected for comparison. The encoding results are presented in Fig. 2e. It was observed that the encoding accuracies of V3 were lower than those of V1 and V2, which may be attributed to its lower signal-to-noise ratio[1]. Significant differences were observed between the accuracies of SCNN and GWP ($p < 10^{-24}$, one-tailed two-sample $t$-test) for all visual areas, with no significant difference between SCNN and GNet ($p > 0.12$, two-tailed two-sample $t$-test) for V2 and V3. For the colorful natural image dataset, we compared the encoding performance of SCNN with CNN and GWP and selected 500 voxels with the highest encoding performance for each subject for comparison. As depicted in Fig. 2f, the accuracies of SCNN were significantly higher than those of CNN ($p < 10^{-36}$, one-tailed two-sample $t$-test) for all subjects. Moreover, SCNN demonstrated comparable results to GNet for subject1 (SCNN higher than GNet, $p = 1.58 \times 10^{-19}$, one-tailed two sample $t$-test) and subject4 (no significant difference, $p = 0.725$, two-tailed two-sample $t$-test).

In general, the encoding results of the natural image datasets suggest that the unsupervised SCNN-based encoding model outperforms traditional GWP and CNN-based models and can even achieve comparable performance with neural networks optimized with brain response as the target.

**Image reconstruction result**. The image reconstruction task aims to reconstruct the images perceived by the participant from the fMRI responses. Based on the pre-trained encoding models, we accomplished this task on handwritten characters, handwritten digits, and colorful natural image datasets. The prior image set for the handwritten character dataset consisted of the images of six characters in the TICH dataset (excluding the test set images). For the handwritten digit dataset, the prior image set comprised 2000 prior handwritten 6 and 9 images. The images in the validation set of ImageNet were used as the prior image set for the colorful natural image dataset. It is noteworthy that only 200 voxels selected from the train set data were utilized for this task. To reconstruct each image in the test set, the top 15 images of the

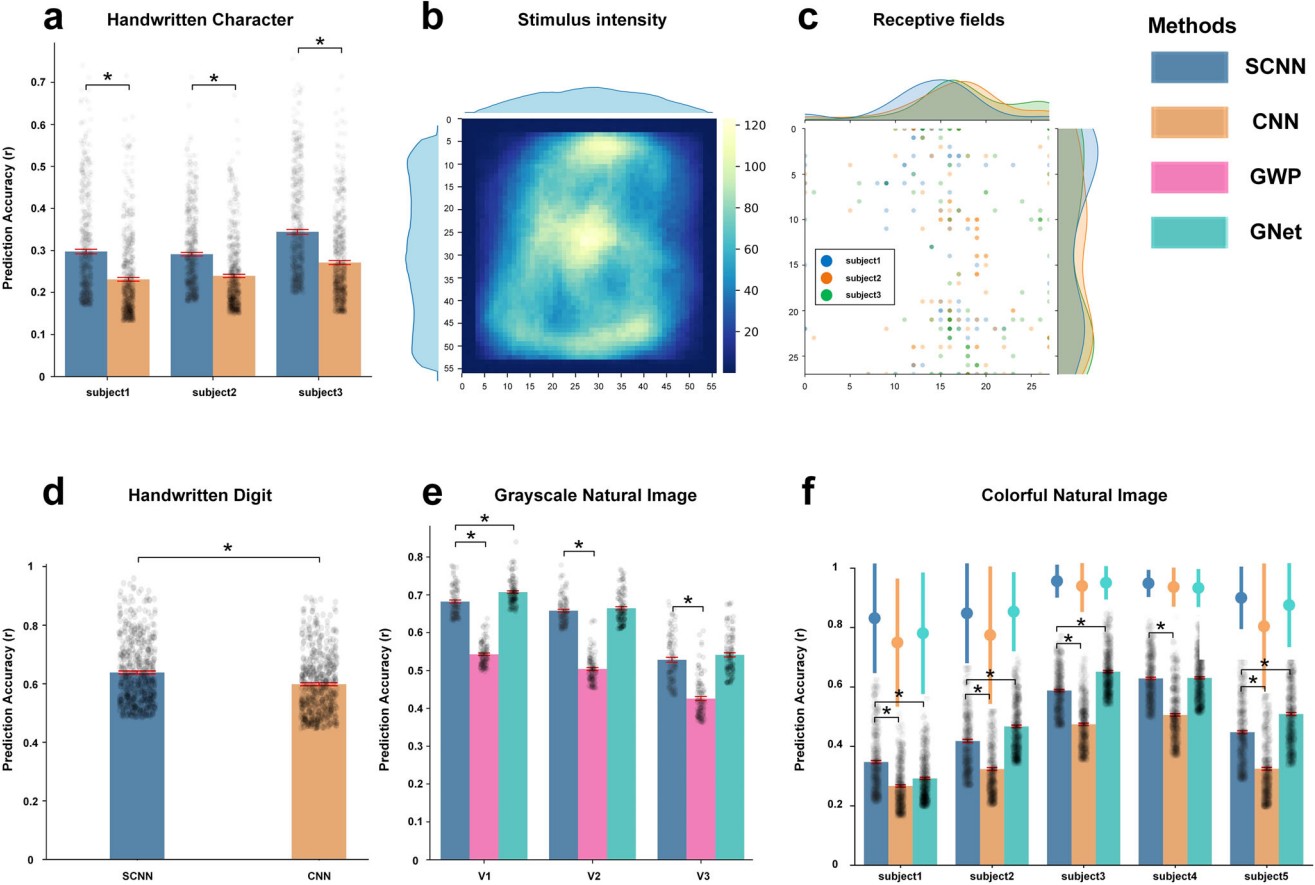

**Fig. 2 The encoding results of the selected voxels in different datasets. a** The encoding accuracies ($n = 500$) of different subjects in the handwritten character dataset. **b** The mean stimulus intensities in the train set of the handwritten character dataset. **c** The receptive field locations of the 100 most predictable voxels of the handwritten character dataset. A smaller transparency represents a larger number of voxels. **d** The encoding accuracies ($n = 500$) of the handwritten digit dataset. **e** The encoding accuracies ($n = 200$) of different visual areas in the grayscale natural image dataset. **f** The encoding accuracies ($n = 500$) and noise ceilings (mean ± standard deviation) of different subjects in the colorful natural image dataset. The bar charts represent the mean ± SEM (standard error of the mean) of the encoding accuracies, and * represents $p < 10^{-12}$ for a one-tailed two-sample t-test.

prior image set with the highest likelihood with observed responses were averaged, resulting in the reconstructed image.

The reconstruction results of the handwritten character dataset demonstrated that our reconstructions can effectively distinguish different characters and can reconstruct images that belong to the same character with different writing styles (see Fig. 3a, b). Similarly, our approach yielded promising reconstruction results on the handwritten digit dataset (see Fig. 3c). The reconstruction results of the colorful natural image dataset are presented in Fig. 3d. Although our model can only deal with grayscale images, which resulted in the loss of color information in the reconstruction results, the reconstructions retained the structural information, such as shape and position, of the original stimuli. Additionally, we observed that the prior images with the highest likelihood exhibited high structural similarities to the real stimuli (see Fig. 3e). The reconstruction results were quantitatively evaluated using PCC and Structural Similarity Index (SSIM)[30] and were compared with other benchmark methods, including CNN, GNet, SMLR[31], DCCAE[32], DGMM+[33], and Denoiser GAN[34]. As presented in Table 2, our approach achieved competitive or superior performance compared to these methods.

**Image identification result**. The image identification task aims to identify the image seen by the participant from the fMRI responses, and this task was accomplished on the grayscale natural image dataset. The encoding model was utilized to generate

predicted fMRI responses for all images in the test set. The images perceived by the participants were identified by matching the measured responses to the predicted responses. As per a previous study[1], 500 voxels with the highest predictive power were employed for this task. Our approach achieved identification accuracies of 96.67% (116/120) and 90.83% (109/120) for the two participants, respectively, which were higher than those of the GWP model (92% and 72%) and GNet (90% and 73.33%). The correlation maps between measured and predicted responses for the two participants are presented in Fig. 4. For most of the rows in the correlation maps, the elements on the diagonal were significantly larger than the others, indicating that our approach exhibited excellent identification ability.

**Effect of hyperparameters on decoding tasks**. The selection of hyper-parameters directly affects the performance of downstream decoding tasks. To evaluate the impact of hyper-parameters on the image reconstruction task, we investigated the reconstruction performance with two hyper-parameters: the number of selected voxels and the number of averaged images. Specifically, we examined the reconstruction performance using 50, 100, 200, and 500 voxels and 1, 5, 10, 15, 20, 25, and 30 images on the handwritten character dataset. As illustrated in Fig. 5a, the PCC index increased with the number of images and reached its peak at the voxel number of 200. Conversely, the SSIM index decreased with the increase in the number of images and reached its peak at the

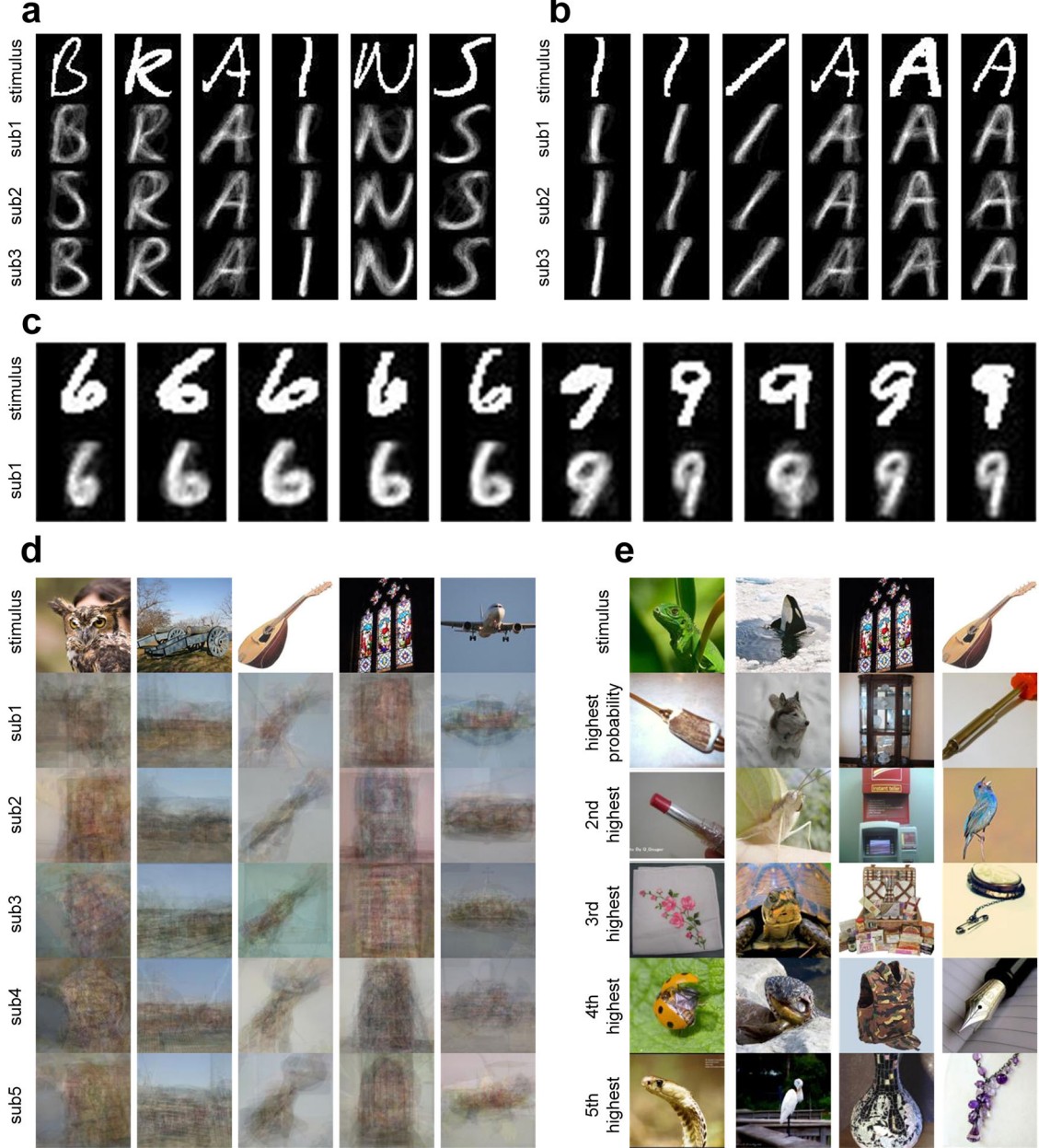

**Fig. 3 The reconstruction results of different datasets. a** The reconstructions of different handwritten characters (B, R, A, I, N, and S). The images in the first row are the presented images (ground truth), and the images in the second to fourth rows are the reconstruction results of the 3 subjects. **b** The reconstructions of the same character with different writing styles. **c** The reconstructions of handwritten digits. The handwritten digit images are adapted from the MNIST database (http://yann.lecun.com/exdb/mnist/) with permission. **d** The reconstructions of natural images. **e** Examples of prior images with the highest likelihoods of the colorful natural image datasets. The colorful natural images in **d** and **e** are adapted from the ImageNet database[52] with permission.

voxel number of 200 and 500. A larger number of voxels contained more stimulus information but also introduced more noise. Similarly, a larger number of images made the reconstruction more realistic but also blurred the reconstruction. To evaluate the impact of hyper-parameters on the image identification task, we investigated the identification accuracies with 100, 500, 1000, and 2000 voxels. As depicted in Fig. 5b, our approach achieved the highest accuracies when 500 voxels were utilized.

**Reproducibility analysis.** In the proposed encoding model, the unsupervised SCNN was utilized to extract features of the visual stimuli, and the training process of SCNN was influenced by its initial values. To investigate the impact of initial values on the

encoding performance, we trained another SCNN with different initial values on the grayscale natural image dataset and compared its encoding performance with the original one. For each subject, the top 500 voxels with the highest encoding performance were selected for comparison, and no significant differences were observed between the two encoding results (subject1: $p = 0.1$, subject2: $p = 0.47$, two-tailed two-sample $t$-test).

## Discussion
In this work, a visual perception encoding model based on SCNN was proposed, comprising the SCNN feature extractor and voxel-wise response predictors. Unlike conventional Gabor and CNN-based methods that employ real-value computation, the proposed

model utilized spike-driven SCNN to process visual information in a more biologically plausible manner. The model demonstrated remarkable success in predicting brain activity evoked by hand-written characters, handwritten digits, and natural images, using a simple two-layer unsupervised SCNN and four publicly available datasets as the test bed. Moreover, promising results were obtained in image reconstruction and identification tasks using our encoding models, suggesting the potential of the model in addressing practical brain-reading problems.

Neural encoding can bridge artificial intelligence models and the human brain. By establishing a linear mapping from model features to brain activity, the similarity of information processing between the model and the brain can be quantitatively evaluated. Therefore, it is reasonable to assume that a model with high biological plausibility is more likely to achieve superior encoding performance. In light of this, we developed an SCNN-based encoding model to predict brain responses elicited by various visual inputs. The SCNN architecture combines the network structure of CNN, which has been shown to be effective for neural encoding[2,4,13,14], with the computational rules of SNN that are more biologically realistic. To extract meaningful visual features, we employed an SCNN consisting of a DoG layer and a con-volutional layer, which simulate information processing in the retina and visual cortex, respectively. Our model outperformed

other benchmark methods (Gabor and CNN-based encoding models), in terms of encoding performance on experimental data, highlighting the superiority of SCNN in visual perception encoding.

Despite its biological plausibility, SCNN simulates information processing at the level of individual neurons, while fMRI measures large-scale brain activity, with each voxel's signal repre-senting the joint activity of a large number of neurons. Therefore, regression models are crucial for voxel-level encoding, as they map the activations of multiple SCNN neurons to the responses of single voxels. Previous studies have demonstrated the neuronal population receptive field properties[35,36] of fMRI data, indicating that each voxel in the visual cortex (especially in V1–3) only receives visual input from a fixed range of the visual field. Based on this theory, we employed a feature selection algorithm that matched the receptive field location for each voxel, which was more consistent with the real visual mechanism and reduced the risk of overfitting.

The question of whether the brain operates under supervised or unsupervised conditions has been a topic of debate. In lieu of utilizing supervised CNNs, we employed an unsupervised SCNN trained via STDP in our model. The findings of this study suggest that the early visual areas of the visual cortex are more inclined to acquire visual representations in an unsupervised manner. Additionally, the STDP-based SCNN offers several advantages in terms of neural encoding. Firstly, it is biologically plausible due to the bioinspired nature of STDP as a learning rule. Secondly, it is capable of handling both labeled and unlabeled data. Lastly, it is particularly well-suited for small sample datasets, such as those obtained via fMRI.

The realization of neural decoding tasks serves as the foun-dation for numerous brain-reading applications, such as BCI[37]. Two types of decoding models exist: those derived from encoding models and those constructed directly in an end-to-end manner. The former offers voxel-level functional descriptions while completing decoding tasks[5]. However, recent breakthroughs in decoding have primarily been achieved using the latter models[33,38,39]. In this study, we successfully completed down-stream decoding tasks, including image reconstruction and identification, based on the encoding model. The results demonstrate that our approach outperformed other end-to-end models in both decoding tasks. This finding further confirms the effectiveness of our encoding model and suggests that encoding-

**Table 2 Reconstruction performance with different methods.**

| Dataset | Methods | PCC | SSIM |
|---|---|---|---|
| Handwritten character | SMLR[31] | 0.481 ± 0.096 | 0.191 ± 0.043 |
| | DGMM+[33] | 0.502 ± 0.193 | 0.360 ± 0.050 |
| | Denoiser GAN[34] | 0.319 ± 0.032 | 0.465 ± 0.031 |
| | CNN | 0.188 ± 0.149 | 0.258 ± 0.073 |
| | SCNN (Ours) | 0.561 ± 0.169 | 0.409 ± 0.166 |
| Handwritten digit | DCCAE-A[32] | 0.548 ± 0.044 | 0.358 ± 0.097 |
| | DCCAE-S[32] | 0.511 ± 0.057 | 0.552 ± 0.088 |
| | Denoiser GAN[34] | 0.531 ± 0.049 | 0.529 ± 0.043 |
| | CNN | 0.738 ± 0.107 | 0.532 ± 0.122 |
| | SCNN (Ours) | 0.762 ± 0.056 | 0.578 ± 0.082 |
| Colorful natural image | CNN | 0.191 ± 0.271 | 0.344 ± 0.120 |
| | GNet | 0.166 ± 0.252 | 0.343 ± 0.109 |
| | SCNN (Ours) | 0.206 ± 0.231 | 0.347 ± 0.116 |

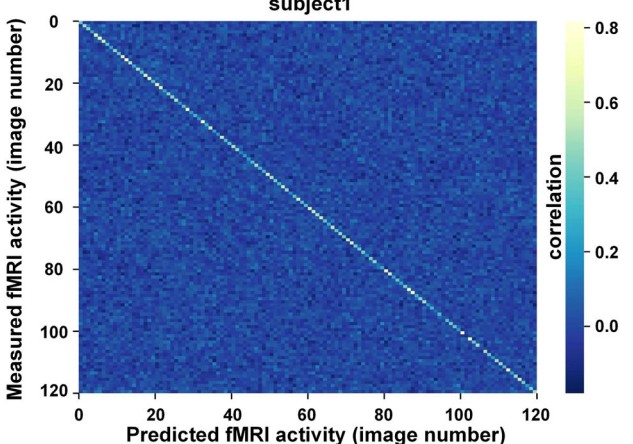

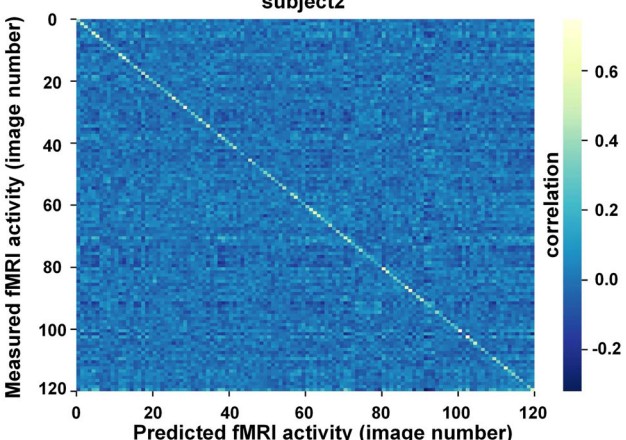

**Fig. 4 Image identification results on natural image dataset.** The correlation maps of the measured and predicted fMRI responses to test set images for the two participants. The element in the $m_{th}$ column and $n_{th}$ row represents the correlation between the measured fMRI response for the $m_{th}$ image and the predicted fMRI response for the $n_{th}$ image.

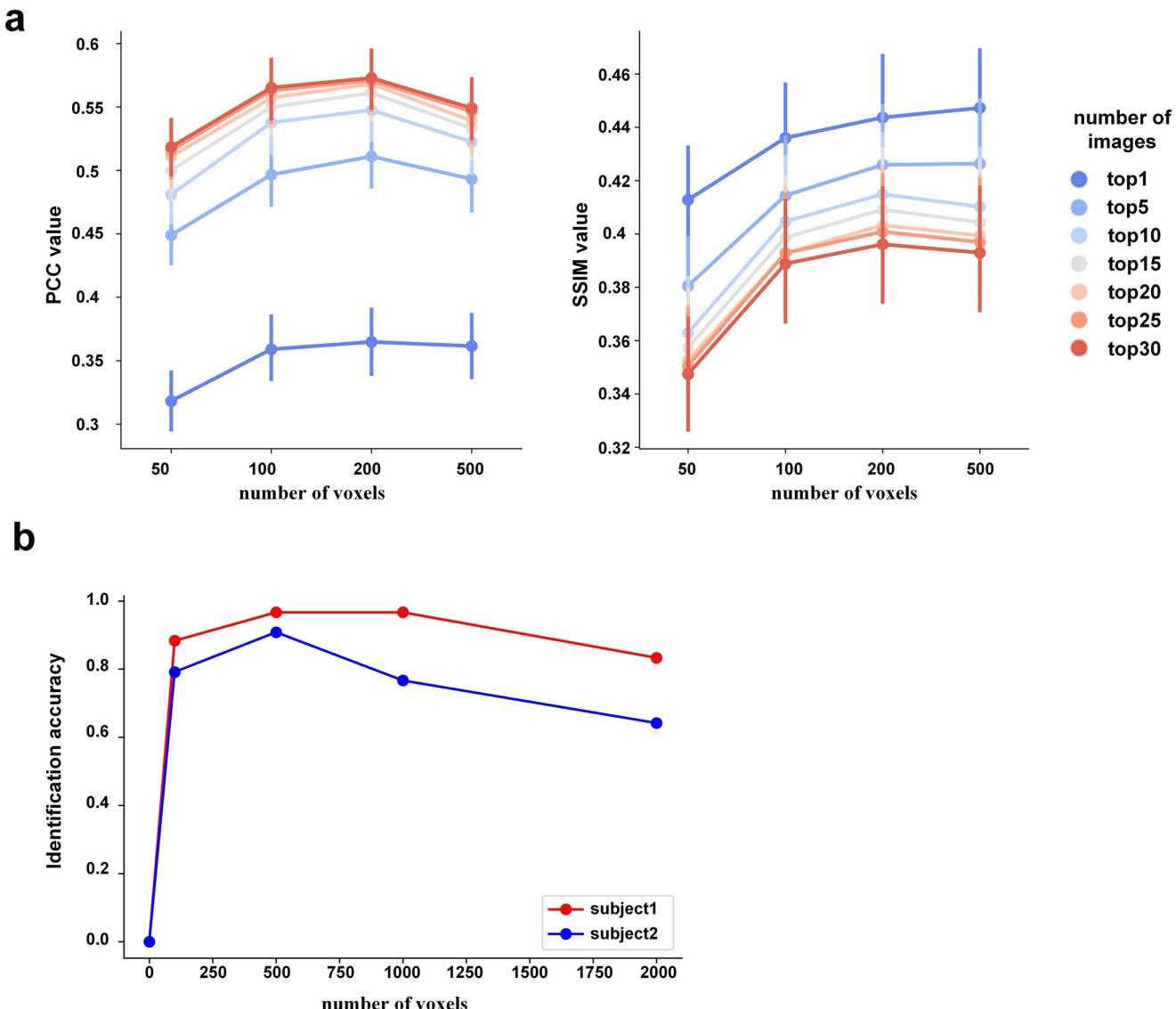

**Fig. 5 The effect of parameters on decoding tasks. a** The reconstruction performance (PCC and SSIM) of different hyper-parameters (number of selected voxels and number of averaged images) on the handwritten character dataset, the dots represent mean values, and the error bars represent 95% confidence intervals. **b** The identification accuracies with different numbers of voxels for the two subjects in the grayscale natural image dataset.

based approaches hold significant potential for solving decoding tasks.

Despite the progress made in neural encoding using SCNN, there remain several limitations. First, the architectures of SNNs are typically shallower than those of deep-learning networks, which restricts their ability to extract complex and hierarchical visual features. Recent studies have attempted to address this issue and have made some headway[23,24,40]. The incorporation of a deeper SCNN into our model would further enhance encoding performance and enable investigation of the hierarchical structure of the visual cortex. Second, the Integrate-and-Fire neuron utilized in our study is a simplification of biological neurons. The use of more realistic neurons, such as leaky Integrate-and-Fire and Hodgkin-Huxley neurons[41], would further enhance the biological plausibility of our encoding model. Third, the parameters of STDP and network architecture were selected from previous works[23,24], and the impact of different parameters on encoding performance requires further exploration.

In conclusion, this work presents a powerful tool for neural encoding. On the one hand, we combined the structure of CNNs and the calculation rules of SNNs to model the visual system and constructed voxel-wise encoding models based on the receptive field mechanism. On the other hand, we demonstrated that our model can be utilized to perform practical decoding tasks, such as image reconstruction and identification. We anticipate that SCNN-based encoding models will provide valuable insights into the visual mechanism and contribute to the resolution of BCI and computer vision tasks. Furthermore, we plan to extend the use of SNNs to encoding tasks of other cognitive functions (e.g., imagination and memory) in the future.

## Methods

**SCNN-based encoding model**. An SCNN-based encoding model was proposed in this study to predict fMRI activities that are elicited by input visual stimuli. The encoding model was comprised of voxel-wise regression models and a SCNN feature extractor. Initially, the unsupervised SCNN was utilized to extract the stimulus features for each input image. Subsequently, linear regression models were constructed to project the SCNN features

into fMRI responses. The architecture of the encoding model is depicted in Fig. 1a.

*SCNN feature extractor.* To extract stimuli features, a simple two-layer SCNN was employed in this study. The first layer, known as the Difference of Gaussians (DoG) layer, was designed to emulate neural processing in retinal ganglion cells[42,43]. The parameter settings for this layer were based on previous research[23,24]. For both handwritten characters and natural images, each input image underwent convolution with six DoG filters with zero padding. ON- and OFF-center DoG filters with sizes of $3 \times 3$, $7 \times 7$, and $13 \times 13$, and standard deviations of $(3/9, 6/9)$, $(7/9, 14/9)$, and $(13/9, 26/9)$ were utilized. The padding size was set to 6 for this study. For handwritten digits, each input image underwent convolution with two DoG filters with zero padding. ON- and OFF-center DoG filters with a size of $7 \times 7$ and standard deviations of $(1, 2)$ were utilized. The padding size was set to 3. Subsequently, DoG features were transformed into spike waves using intensity-to-latency encoding[44] with a length of 30. Specifically, DoG feature values greater than 50 were sorted in descending order and equally distributed into 30 bins to generate the spike waves. Prior to being passed to the next layer, the output spikes underwent max pooling with a window size of $2 \times 2$ and a stride of 2.

The second layer of the SCNN corresponds to the convolutional layer, which was designed to emulate the information integration mechanism of the visual cortex. In this layer, 64 convolutional kernels comprised of Integrate-and-Fire (IF) neurons were utilized to process the input spikes. The window size of the convolutional kernels was $5 \times 5$, and the padding size was 2. Each IF neuron gathered input spikes from its receptive field and emitted a spike when its voltage reached the threshold. This can be expressed mathematically as follows:

$$v_i(t) = v_i(t-1) + \sum_j w_{ij} \times s_j(t-1), \qquad (1)$$

$$s_i(t) = 1 \text{ and } v_i(t) = 0, \text{ if } v_i(t) \geq v_{th}, \qquad (2)$$

where $v_i(t)$ represents the voltage of the $i_{th}$ IF neuron at time step t, while $w_{ij}$ signifies the synaptic weight between the $i_{th}$ neuron and the $j_{th}$ input spikes within the neuron's receptive field. The firing threshold, denoted by $v_{th}$, is set at 10. For each image, neurons are permitted to fire a maximum of once. The inhibition mechanism is employed in the convolutional layer, allowing only the neuron with the earliest spike time to fire at each position in the feature maps. Synaptic weights are updated through Spike-Timing-Dependent Plasticity (STDP), which can be expressed as:

$$\Delta w_{ij} = \begin{cases} a^+ \times w_{ij} \times \left(1 - w_{ij}\right), & \text{if } t_j - t_i \leq 0, \\ a^- \times w_{ij} \times \left(1 - w_{ij}\right), & \text{if } t_j - t_i > 0, \end{cases} \qquad (3)$$

where $\Delta w_{ij}$ denotes the weight modification, $a^+$ and $a^-$ represent the learning rates (set at 0.004 and $-0.003$, respectively)[23], and $t_i$ and $t_j$ indicate the spike times of the $i_{th}$ neuron and $j_{th}$ input spikes, respectively. The learning convergence, as defined by Kheradpisheh et al.[23], is calculated using the following equation:

$$C = \sum_i \sum_j w_{ij} \times (1 - w_{ij})/N, \qquad (4)$$

where $N$ represents the total number of synaptic weights. The training of the convolutional layer ceases when C is below 0.01. The SCNN implementation is based on the SpykeTorch platform[45]. After training the SCNN, the firing threshold $v_{th}$ is set to infinity, and the voltage value at the final time step in each neuron is measured as the SCNN feature of the visual stimuli. As

the voltages in the convolutional neurons accumulate over time and are never reset when $v_{th}$ is infinite, the final voltage values (SCNN feature) reflect the SCNN's activation in response to the visual stimuli.

*Responses prediction algorithm.* With the obtained SCNN feature $F \in \mathscr{R}^{64 \times h \times w}$, a linear regression model is constructed for each voxel to predict the fMRI response Y. To avoid the overfitting problem, the receptive field mechanism is introduced into the regression models, where each voxel only receives the input at a specific location of the SCNN feature map. To identify the optimal receptive field location for each voxel (different voxels can have the same preferred receptive field), all locations on the SCNN feature maps are examined to fit the regression model, and threefold cross-validation is performed on the training data. The regression model's expression and objective function are defined as:

$$y_v = w \times f_{ij} + \epsilon, \qquad (5)$$

$$\min_w ||w \times f_{ij} - y_v||_2^2, \qquad (6)$$

where $y_v$ represents the fMRI response of voxel $v$, $w$ denotes the weight parameters in the regression model and $f_{ij} \in \mathscr{R}^{64 \times 1}$ ($i = 1, 2, \ldots, h, j = 1, 2, \ldots, w$) signifies the feature vector at location $(i, j)$ of the SCNN feature maps. The regression accuracy is quantified using the coefficient of determination ($R^2$) of the predicted and observed responses, and the feature location with the highest $R^2$ is chosen as the receptive field location for each voxel. Lastly, the regression model for each voxel is retrained on the entire training data based on the determined receptive field location.

**Downstream decoding tasks.** Two downstream decoding tasks were performed based on the encoding models, namely image reconstruction and image identification. The objective of the image reconstruction task is to reconstruct the perceived image from the observed fMRI response, while the image identification task aims to determine the image that was viewed. The specific methodologies employed for these tasks are expounded upon as follows.

*Image reconstruction.* As depicted in Fig. 1b, the image reconstruction task was executed by utilizing an extensive prior image set. Initially, the encoding model was employed to generate the anticipated fMRI responses for all images in the prior image set. Subsequently, the likelihood of the observed fMRI response r given the prior image s was estimated, which can be mathematically represented as a multivariate Gaussian distribution:

$$p(r|s) \propto \exp\left\{-(r - \hat{r}(s))\sum^{-1}(r - \hat{r}(s))'\right\} \qquad (7)$$

$$\sum = \text{cov}(r - \hat{r}(s)) \qquad (8)$$

Where $\hat{r}(s)$ represents the predicted fMRI response of s, and $\Sigma$ signifies the noise covariance matrix for train samples. Finally, the prior images that elicited the highest likelihood of evoking the observed fMRI response were averaged to derive the reconstruction result.

*Image identification.* Figure 1c illustrates the methodology employed for the image identification task. The test set images were fed into the encoding model to generate the predicted fMRI responses. Subsequently, the Pearson's correlation coefficients (PCCs) between the predicted fMRI responses and the observed fMRI response were computed. The image that exhibited the highest correlation between its predicted fMRI response and the

observed response was deemed to be the image viewed by the subject.

**fMRI datasets**. To validate the encoding model, four publicly available datasets that have been extensively utilized in prior research[1,25–27,33,38,46] were utilized, namely the handwritten character, handwritten digits, grayscale natural image, and colorful natural image datasets. The fundamental characteristics of these datasets are presented in Table 1, and a brief overview of each dataset is provided below.

*Handwritten character dataset*. This dataset comprises fMRI data obtained from three participants as they viewed handwritten character images. A total of 360 images depicting 6 characters (B, R, A, I, N, and S) with the size of $56 \times 56$ were presented to each participant, sourced from the TICH character dataset[47]. A white square was added to each image as a fixation point. During the experiment, each image was displayed for 1 s (flashed at 2.5 Hz), followed by a 3-s black background, and 3 T fMRI data were simultaneously collected (TR = 1.74 s, voxel size = $2 \times 2 \times 2\,mm^3$). The voxel-level fMRI responses of visual areas V1 and V2 for each visual stimulus were estimated using general linear models[48]. The same train/test set split as the original work[25] was adopted, which comprised 270 and 90 class-balanced examples, respectively.

*Handwritten digit dataset*. This dataset comprises fMRI data obtained from one participant while viewing handwritten digit images[26]. During the experiment, 100 handwritten 6 and 9 images with the size of $28 \times 28$ were presented to the participant, with each image displayed for 12.5 s and flashed at 6 Hz. The fMRI responses of V1, V2, and V3 were captured using a Siemens 3 T MRI system (TR = 2.5 s, voxel size = $2 \times 2 \times 2\,mm^3$). The train and test sets comprised 90 and 10 examples, respectively. Additionally, this dataset provided 2000 prior handwritten 6 and 9 images that were not utilized in the fMRI experiment for the image reconstruction task.

*Grayscale natural image dataset*. This dataset comprises fMRI data obtained from two participants as they viewed grayscale natural images[1]. The experiment was divided into train and test stages. During the training stage, the participants were presented with 1750 images, each of which was displayed for a duration of 1 s (flashed at 2 Hz), followed by a 3 s gray background. In the test stage, the participants were shown 120 images that were distinct from the ones used in the training stage. The fMRI data was acquired simultaneously in both stages of the experiment using a 3 T scanner (TR = 1 s, voxel size = $2 \times 2 \times 2.5\,mm^3$). The voxel-level fMRI responses of visual areas V1–V3 were estimated for each visual stimulus. To mitigate computational complexity, the natural images were downsampled from $500 \times 500$ to $128 \times 128$ pixels.

*Colorful natural image dataset*. This dataset comprises fMRI data obtained from five participants as they viewed colorful natural images[27]. The experiment consisted of two sessions, namely the training image session and the test image session. During the training image session, each participant was presented with 1200 images from 150 categories, with each image being displayed only once (flashed at 2 Hz for 9 s). In the test image session, each participant was shown 50 images from 50 categories, with each image being presented 35 times. The fMRI responses of multiple visual areas on the ventral visual pathway were collected using a 3 T Siemens scanner (TR = 3 s, voxel size = $3 \times 3 \times 3\,mm^3$), and V1, V2, and V3 were selected as regions of interest for this study.

Prior to being fed into the SCNN, the natural images were converted from RGB format to grayscale format and downsampled from $500 \times 500$ to $128 \times 128$ pixels.

**Noise ceiling estimation**. The encoding accuracies of the colorful natural image dataset were compared with noise ceilings, which represent the upper limit of the accuracies in the presence of noise. To calculate the noise ceiling for each voxel, we employed a method that has been commonly used in previous studies[13,49–51]. This method assumes that the noise follows a Gaussian distribution with a mean of zero and that the observed fMRI signal is equal to the response plus noise. Initially, we estimated the standard deviation of the noise $\hat{\sigma}_N$ using the following formula:

$$\hat{\sigma}_N = \sqrt{\mathrm{mean}(\sigma_R^2)}, \qquad (9)$$

Where $\sigma_R^2$ represents the variance of the responses across 35 repeated sessions of each test image. Subsequently, we calculated the variance of the response by subtracting the variance of the noise from the variance of the mean response:

$$\hat{\sigma}_R^2 = \mathrm{var}(\mu_R) - \hat{\sigma}_N^2, \qquad (10)$$

Where $\mu_R$ represents the mean responses across the repeated sessions of each test image. Finally, we drew samples from the response and noise distributions to obtain their simulations and generated the simulated signal by summing the simulated response and noise. We conducted 1000 simulations and calculated the PCC between the simulated signal and response in each simulation. The mean PCC value was taken as the noise ceiling.

**Statistics and reproducibility**. In Fig. 2, we performed a one-tailed two-sample $t$-test to compare the encoding accuracies of different methods on each dataset, and the sample sizes were described in figure captions. In reproducibility analysis, we conducted a two-tailed two-sample $t$-test to estimate whether the encoding accuracies ($n = 500$) between the SCNNs with different initial values exhibited any significant statistical differences; the corresponding $p$-values were reported in the "Results" section.

**Reporting summary**. Further information on research design is available in the Nature Portfolio Reporting Summary linked to this article.

### Data availability

The handwritten character dataset is publicly available at http://sciencesanne.com/research/, the handwritten digit dataset is publicly available at http://hdl.handle.net/11633/di.dcc.DSC_2018.00112_485, the grayscale natural image dataset is publicly available at https://crcns.org/datasets/vc/vim-1, the colorful natural image dataset is publicly available at https://github.com/KamitaniLab/GenericObjectDecoding. The source data underlying Figs. 2, 4, and 5 can be found in Supplementary Data 1, 2, 3.

### Code availability

The code that supports the findings of this study is available from https://github.com/wang1239435478/Neural-encoding-with-unsupervised-spiking-convolutional-spiking-neural-networks.

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

## Acknowledgements

This work was supported by the STI 2030-Major Projects 2022ZD0208900 and the National Natural Science Foundation of China (Nos. 82121003, 62036003, 62276051, and 82072006), Medical-Engineering Cooperation Funds from University of Electronic Science and Technology of China (ZYGX2021YGLH201), Innovation Team and Talents Cultivation Program of National Administration of Traditional Chinese Medicine (No. ZYYCXTD-D-202003).

## Author contributions

Chong Wang designed the project and wrote the paper; Yuting Wang, Yun-Shuang Fan, and Ting Zou prepared the data; Wei Huang, Wei Sheng, and Tao Liu analyzed data and built models; Hongmei Yan, Rong Li, and Huafu Chen supervised the project and revised the paper.

## Competing interests

The authors declare no competing interests.
