## [Peer Review File · Communications Biology]

Reviewers' comments:

Reviewer #1 (Remarks to the Author):

CNNs are widely used computational tools for modeling neural encoding and decoding mechanisms in the brain, but their units are vastly simpler than real neurons. The extent to which networks based on such simple units accurately reflect the principles of computations implemented in the brain is a question of long-standing interest. In this paper, the authors made a spiking neural network to predict measured fMRI responses on the basis of what visual stimuli were shown. They tested their network on image reconstruction task (using Kay et al image set) and an image identification task (using Van der Maaten BRAINS character set), and found that their results compare favorably with those of previously published networks.

The claim that networks that incorporate spike timing out perform networks of units with continuously variable activity is surprising, and would have important implications for the field if it proves to be a consistent finding.

One reason for caution in accepting these results at face value is that the model under discussion was tested on only two image sets, which were presumably also used for conceiving and developing the model. Thus we could be looking here at specific testing conditions under which the current model enjoys a particular advantage.

A more convincing test of the generality of the spiking model advantage would be to repeat the same tests other suitable data sets. For instance, the Miyawaki et al 2008 and the Van Gerven, et al, Neural computation, 2010 would be a good comparison for the letter ID task. The authors could repeat the image reconstruction experiment on the NSD dataset of Allen et al 2022, in essence competing with the GNet model on it's home turf.

Reviewer #2 (Remarks to the Author):

In their manuscript the authors introduce an unsupervised spiking convolutional neural network and test its neural predictivity on two available fMRI datasets. The proposed network showed improved or similar encoding performance compared to control networks. Moreover, the network can be used to reconstruct or identify the stimulus which evoked a particular fMRI response. The paper is concise, the topic is timely given the biological implausibility of learning and information transmission in standard CNN approaches, and the methods and results appear appropriate. I have several comments and questions/suggestions to improve the paper which I will lay out in the following.

1) For the prediction accuracy in Fig. 2a and 3, it would be helpful to see some sort of noise ceiling. That is how well is the prediction accuracy compared to the maximum achievable prediction accuracy given the noise present in the data. There are different ways to compute such noise ceilings and the best way might depend on the data. But in this case, it could be useful to build brain-to-brain encoding models to test how well each subject's fMRI data can encode another subject's fMRI data. Another way could be to compute the split-half correlation (i.e., splitting the voxel's responses for each condition into half and correlating them) of the selected voxels in the fMRI data.

2) While biological plausibility is a great goal, it is not well motivated in the manuscript what we can gain from using these models beyond higher biological plausibility. Are there certain problems or

neural phenomena that CNNs do not show, but that might emerge in SCNNs? This would be evidence that neural spiking properties are required to capture these aspects of neural processing. Generally, the use of SCNNs appear more useful at the level of predicting individual neurons where the current abstractions in CNNs might not be sufficient. It would be helpful to better motivate and discuss these potentials of SCNNs in the paper.

3) More regarding biological plausibility: The spiking property of neurons seems clearly more biologically plausible than in standard CNNs, however, the approach that activation simply accumulated in the convolutional layer over time, does not seem biologically plausible either. How do the authors argue for this implementation and are there current alternatives? These constraints given the proposed model should be discussed.

4) Some details and motivation about the receptive field-based feature algorithm are unclear or missing. How critical is this step for the predictive performance of the SCNN? Has the same been done for the CNN (e.g. Fig. 2)? Does this mean that each voxel's encoding model is trained on only a tiny part of the visual input? Can two voxels have the same preferred receptive field and if so, why are there no overlaps between voxels and/or subjects in Fig. 2b? Do the authors expect this would also work on higher level visual areas, or is this particularly useful for early visual cortex areas? Please clarify and motivate this methodological step further.

5) Several sections in the manuscript are unclear and some more details and conjunctions between sentences would be helpful for the reader. For example, in line 50, the sentence about representational similarity analysis comes very sudden and the "therefore" in the next sentence implies causality, that seems too strong. Generally, it might not be clear to all readers what RSA is and what the link between RSA and encoding models is. Also in the Results section, it would help to motivate or repeat the question/problem that is being addressed in this section.

6) The figures need to be improved. There is no axis label or color bar of the intensity plot in Figure 2b. Further, the x-label in Figure 5b is missing. And in Figure 5a, instead of 'correlation (r)', it would be helpful to use 'Subject 1' and 'Subject 2' as subtitle and move 'correlation' vertically next to color bar. Moreover, the description of error bars is missing in the legends of the plots.

Minor comments:

- 1) There are many typos / grammatical errors in the manuscript that need to be fixed.
- 2) Abbreviation PCCs in line 88 has not yet been introduced.

Reviewer #3 (Remarks to the Author):

The paper addresses neural encoding (prediction of brain response to external stimuli) using unsupervised spiking convolutional neural networks (SCNNs).

The results shown in the paper are relevant for the field, and demonstrate the suitability of the SCNNs for all the tasks analyzed in the paper, which include encoding in symbolic (handwritten characters) and natural images, image reconstruction and identification.

The statistical analysis seems appropriate.

One major concern remains regarding reproducibility. The involved methods involve the use of STDP in convolutional neurons. This algorithm, as any (especially unsupervised) ML approach, may highly

depend on initial values. Also, STDP has several parameters (which modulate the long-term potentiation/depression). Also, the SCNN shown contains an important first layer constituted by Difference of Gaussians responses. The authors do not disclose their choices for the parameters of these methods, which are key for research reproduction. The level of detail provided is insufficient for reproducibility.

University of Electronic Science and Technology of China

Huafu Chen, PhD

Director, MOE Key Lab for Neuroinformation
The Clinical Hospital of Chengdu Brain Science Institute
School of Life Science and Technology
University of Electronic Science and Technology of China
Chengdu 610054, P.R. China
Tel: 013808003171
E-mail: chenhf@uestc.edu.cn

RE: COMMSBIO-23-0421

June 18, 2023

Reviewer 1

REVIEWER COMMENTS:

CNNs are widely used computational tools for modeling neural encoding and decoding mechanisms in the brain, but their units are vastly simpler than real neurons. The extent to which networks based on such simple units accurately reflect the principles of computations implemented in the brain is a question of long-standing interest. In this paper, the authors made a spiking neural network to predict measured fMRI responses on the basis of what visual stimuli were shown. They tested their network on image reconstruction task (using Kay et al image set) and an image identification task (using Van der Maaten BRAINS character set), and found that their results compare favorably with those of previously published networks.

The claim that networks that incorporate spike timing outperform networks of units with continuously variable activity is surprising, and would have important implications for the field if it proves to be a consistent finding.

AUTHOR RESPONSE:

We would like to thank the Reviewer for the positive comments. We have now revised the paper substantially and we believe these updates substantially improve the

manuscript.

REVIEWER COMMENTS:

(1) One reason for caution in accepting these results at face value is that the model under discussion was tested on only two image sets, which were presumably also used for conceiving and developing the model. Thus we could be looking here at specific testing conditions under which the current model enjoys a particular advantage.

A more convincing test of the generality of the spiking model advantage would be to repeat the same tests on other suitable data sets. For instance, the Miyawaki et al 2008 and the Van Gerven, et al, Neural computation, 2010 would be a good comparison for the letter ID task. The authors could repeat the image reconstruction experiment on the NSD dataset of Allen et al 2022, in essence competing with the GNet model on its home turf.

AUTHOR RESPONSE:

We thank the Reviewer for this comment. To tackle this problem, we have repeated the encoding tasks on two other datasets. For the letter ID task, we supplemented the results of the handwritten digit dataset¹ (Neural computation, 2010). For the natural image task, we supplemented the results of the colorful natural image dataset², which allow us to compare SCNN with both CNN and GNet. The encoding results of these two datasets are presented in Fig. 2d, f. The detailed description can be found in the section **Encoding performance on handwritten digit dataset** (page 4, line 21-29): “To verify the encoding performance of the proposed approach on handwritten digit stimuli, we trained the SCNN using 2000 prior images that were not utilized in the fMRI experiment. Voxel-wise encoding models were then constructed on the train set of this dataset. Similarly, CNN-based encoding models were built on the handwritten digit dataset, and the top 500 voxels with the highest encoding performance were selected for comparison. The encoding results are presented in Fig. 2d, and the results indicate that the encoding accuracies of SCNN were significantly higher than those of CNN ($p = 6.78 \times 10^{-18}$, one-tailed two sample t-test)” and **Encoding performance on natural image datasets** (page 5, line 19-26): “For the colorful natural image dataset, we compared the encoding performance of SCNN with CNN and GWP, and selected 500 voxels with the highest encoding performance for each subject for comparison. As depicted in Fig. 2f, the accuracies of SCNN were significantly higher than those of CNN ($p < 10^{-36}$, one-tailed two sample t-test) for all subjects. Moreover, SCNN demonstrated comparable results to GNet for subject1 (SCNN higher than GNet, $p = 1.58 \times 10^{-19}$, one-tailed two sample t-test) and subject4 (no significant difference, $p=0.725$, two-tailed two sample t-test)” in **Results**. In general, the unsupervised SCNN outperformed than task-optimized CNN on both of the two datasets and achieved comparable performance with brain-optimized GNet.

Fig. 2. The encoding results of the selected voxels in different datasets. **a** The encoding accuracies of different subjects in the handwritten character dataset. **b** The mean stimulus intensities in the train set of the handwritten character dataset. **c** The receptive field locations of the 100 most predictable voxels of the handwritten character dataset. A smaller transparency represents a larger number of voxels. **d** The encoding accuracies of the handwritten digit dataset. **e** The encoding accuracies of different visual areas in the grayscale natural image dataset. **f** The encoding accuracies and noise ceilings (mean \pm standard deviation) of different subjects in the colorful natural image dataset. The bar charts represent the mean \pm SEM (standard error of the mean) of the encoding accuracies and * represents $p < 10^{-12}$ for one-tailed two sample t-test.

Moreover, we accomplished the image reconstruction task on these two datasets and made comparison of different methods. The reconstruction method utilized in the revised manuscript refers to a previous study³. To reconstruct each test image, we averaged the prior images that elicited the highest likelihood of evoking the observed fMRI response. A detailed description of the methodology can be found in page 13, line 22-33. As shown in Fig. 3, the reconstructions retained the structural information, such as shape and position, of the original stimuli. Using PCC and SSIM as the quantitative indices, our approach achieves superior reconstruction performance than other benchmark methods (see Table 2).

Fig. 3. The reconstruction results of different datasets. **a** The reconstructions of different handwritten characters (B, R, A, I, N, and S). The images in the first row are the presented images (ground truth), and the images in the second to fourth rows are the reconstruction results of the 3 subjects. **b** The reconstructions of the same character with different writing styles. **c** The reconstructions of handwritten digits. **d** The reconstructions of natural images. **e** Examples of prior images with highest likelihoods of the colorful natural image datasets.

Table 2

Reconstruction performance with different methods.

Dataset	Methods	PCC	SSIM
Handwritten Character	SMLR ⁴	0.481±0.096	0.191±0.043
	DGMM+ ⁵	0.502±0.193	0.360±0.050
	Denoiser GAN ⁶	0.319±0.032	0.465±0.031
	CNN	0.188±0.149	0.258±0.073
	SCNN (Ours)	0.561±0.169	0.409±0.166
Handwritten Digit	DCCAЕ-A ⁷	0.548±0.044	0.358±0.097
	DCCAЕ-S ⁷	0.511±0.057	0.552±0.088
	Denoiser GAN ⁶	0.531±0.049	0.529±0.043
	CNN	0.738±0.107	0.532±0.122
	SCNN (Ours)	0.762±0.056	0.578±0.082
Colorful Natural Image	CNN	0.191±0.271	0.344±0.120
	GNet	0.166±0.252	0.343±0.109
	SCNN (Ours)	0.206±0.231	0.347±0.116

Reviewer 2**REVIEWER COMMENTS:**

In their manuscript the authors introduce an unsupervised spiking convolutional neural network and test its neural predictivity on two available fMRI datasets. The proposed network showed improved or similar encoding performance compared to control networks. Moreover, the network can be used to reconstruct or identify the stimulus which evoked a particular fMRI response. The paper is concise, the topic is timely given the biological implausibility of learning and information transmission in standard CNN approaches, and the methods and results appear appropriate. I have several comments and questions/suggestions to improve the paper which I will lay out in the following.

AUTHOR RESPONSE:

We thank the Reviewer for the positive and constructive suggestions regarding our paper. We have now further revised our paper according to Reviewer’s suggestions, and point-by-point responses can be seen below.

REVIEWER COMMENTS:

(1) For the prediction accuracy in Fig. 2a and 3, it would be helpful to see some sort of noise ceiling. That is how well is the prediction accuracy compared to the maximum achievable prediction accuracy given the noise present in the data. There are different ways to compute such noise ceilings and the best way might depend on the data. But in this case, it could be useful to build brain-to-brain encoding models to test how well each subject’s fMRI data can encode another subject’s fMRI data. Another way could be to compute the split-half correlation (i.e., splitting the voxel’s responses for each condition into half and correlating them) of the selected voxels in the fMRI data.

AUTHOR RESPONSE:

We thank the Reviewer for the valuable comment. As suggested by the Reviewer, the estimation of noise ceiling allow us to evaluate how well is the encoding accuracy. Due to the individual differences between subjects, our encoding models are only applicable to single voxels of single subjects. Moreover, the encoding is based on SCNN features rather than stimulus conditions. In our revised manuscript, we adopted an estimation method of noise ceiling that has been commonly used in previous neural encoding studies⁸⁻¹¹. The estimation of the noise ceiling requires the measurement of the fMRI responses in multiple repetitions for each stimulus, while the datasets utilized in the original manuscript do not provide corresponding information. We supplemented the noise ceiling results on the colorful natural image dataset² (see Fig. 2f), which contains 35 repeated sessions for each visual stimulus in the test set. The method to estimate noise ceiling is described in page 15, line 4-21:

“Noise ceiling estimation. The encoding accuracies of the colorful natural image dataset were compared with noise ceilings, which represent the upper limit of the accuracies in the presence of noise. To calculate the noise ceiling for each voxel, we employed a method that has been commonly used in previous studies⁸⁻¹¹. This method assumes that the noise follows a Gaussian distribution with a mean of zero, and that the observed fMRI signal is equal to the response plus noise. Initially, we estimated the standard deviation of the noise $\hat{\sigma}_N$ using the following formula:

$$\hat{\sigma}_N = \sqrt{\text{mean}(\sigma_R^2)}, \quad (9)$$

Where σ_R^2 represents the variance of the responses across 35 repeated sessions of each test image. Subsequently, we calculated the variance of the response by subtracting the variance of the noise from the variance of the mean response:

$$\hat{\sigma}_R^2 = \text{var}(\mu_R) - \hat{\sigma}_N^2, \quad (10)$$

Where μ_R represents the mean responses across the repeated sessions of each test image. Finally, we drew samples from the response and noise distributions to obtain their simulations, and generated the simulated signal by summing the simulated response and noise. We conducted 1000 simulations and calculated the PCC between the simulated signal and response in each simulation. The mean PCC value was taken as the noise ceiling.”

f**Colorful Natural Image**
Fig. 2f. The encoding accuracies and noise ceilings (mean \pm standard deviation) of different subjects in the colorful natural image dataset. The bar charts represent the mean \pm SEM (standard error of the mean) of the encoding accuracies and * represents $p < 10^{-12}$ for one-tailed two sample t-test.

REVIEWER COMMENTS:

(2) While biological plausibility is a great goal, it is not well motivated in the manuscript what we can gain from using these models beyond higher biological plausibility. Are there certain problems or neural phenomena that CNNs do not show, but that might emerge in SCNNs? This would be evidence that neural spiking properties are required to capture these aspects of neural processing. Generally, the use of SCNNs appear more useful at the level of predicting individual neurons where the current abstractions in CNNs might not be sufficient. It would be helpful to better motivate and discuss these potentials of SCNNs in the paper.

AUTHOR RESPONSE:

We thank the Reviewer for this constructive comment. As a study of neural encoding, the straightforward motivation of the biological plausibility is to obtain higher encoding performance. We have stressed this point in the discussion section of the revised paper (page 10, line 7-15):

“Neural encoding can bridge artificial intelligence models and the human brain. By establishing a linear mapping from model features to brain activity, the similarity of information processing between the model and the brain can be quantitatively evaluated.

Therefore, it is reasonable to assume that a model with high biological plausibility is more likely to achieve superior encoding performance. In light of this, we developed an SCNN-based encoding model to predict brain responses elicited by various visual inputs. The SCNN architecture combines the network structure of CNN, which has been shown to be effective for neural encoding^{10,12-14}, with the computational rules of SNN that are more biologically realistic.”

As mentioned by the Reviewer, the use of SCNNs appear more useful at the level of predicting individual neurons. We have added the discussion for this point in page 10, line 21-25:

“Despite its biological plausibility, SCNN simulates information processing at the level of individual neurons, while fMRI measures large-scale brain activity, with each voxel’s signal representing the joint activity of a large number of neurons. Therefore, regression models are crucial for voxel-level encoding, as they map the activations of multiple SCNN neurons to the responses of single voxels.”

REVIEWER COMMENTS:

(3) More regarding biological plausibility: The spiking property of neurons seems clearly more biologically plausible than in standard CNNs, however, the approach that activation simply accumulated in the convolutional layer over time, does not seem biologically plausible either. How do the authors argue for this implementation and are there current alternatives? These constraints given the proposed model should be discussed.

AUTHOR RESPONSE:

We thank the Reviewer for pointing out this important issue. Although SNNs have the advantage of biologically plausible than traditional CNNs, the Integrate-and-Fire neuron utilized in our model is only a simplification of realistic neurons. There are still approaches to further enhance the biologically plausible. We have discussed this point in page 11, line 13-16:

“Second, the Integrate-and-Fire neuron utilized in our study is a simplification of biological neurons. The use of more realistic neurons, such as leaky Integrate-and-Fire and Hodgkin-Huxley neurons¹⁵, would further enhance the biological plausibility of our encoding model.”

REVIEWER COMMENTS:

(4) Some details and motivation about the receptive field-based feature algorithm are unclear or missing. How critical is this step for the predictive performance of the SCNN? Has the same been done for the CNN (e.g. Fig. 2)? Does this mean that each voxel’s encoding model is trained on only a tiny part of the visual input? Can two voxels have the same preferred receptive field and if so, why are there no overlaps between voxels and/or subjects in Fig. 2b? Do the authors expect this would also work on higher level visual areas, or is this particularly useful for early visual cortex areas? Please clarify and motivate this methodological step further.

AUTHOR RESPONSE:

We thank the Reviewer for this comment. We would like to answer the Reviewer's questions first. The importance of the receptive field algorithm can be summarized into two main points: 1. it is consistent with the real visual mechanism^{16,17}; 2. it can reduce the risk of overfitting (for example, the feature dimension of grayscale natural image dataset can be reduced from 262144 to 64 using this algorithm, and its training set sample size is only 1750). To make the result comparable, the same algorithm was adopted for CNN-based encoding models. In the receptive field algorithm, each voxel only receives the input at a specific location of the SCNN feature map, and its proportion is $1/hw$ of the original feature. Different voxels can have the same preferred receptive field. To reflect this point, we have added the transparency to Fig. 2c, and a smaller transparency represents a larger number of voxels. We believe that this algorithm is also applicable to higher level visual areas, but on deeper convolutional layers, as higher level visual areas have much larger receptive fields on original visual stimuli.

Fig. 2c. The receptive field locations of the 100 most predictable voxels of the handwritten character dataset. A smaller transparency represents a larger number of voxels.

We have completed these details in the **Methods** (page 12, line 41-42 and page 13, line 1-4) and **Discussion** (page 10, line 25-30) sections of the revised paper:

“To avoid overfitting problem, the receptive field mechanism is introduced into the regression models, where each voxel only receives the input at a specific location of the SCNN feature map. To identify the optimal receptive field location for each voxel (different voxels can have the same preferred receptive field), all locations on the SCNN feature maps are examined to fit the regression model, and a 3-fold cross-validation is performed on the training data.”

“Previous studies have demonstrated the neuronal population receptive field properties^{16,17} of fMRI data, indicating that each voxel in the visual cortex (especially in V1-3) only receives visual input from a fixed range of the visual field. Based on this theory, we employed a feature selection algorithm that matched the receptive field location for each voxel, which was more consistent with the real visual mechanism and reduced the risk of overfitting.”

REVIEWER COMMENTS:

(5) Several sections in the manuscript are unclear and some more details and conjunctions between sentences would be helpful for the reader. For example, in line 50, the sentence about representational similarity analysis comes very sudden and the “therefore” in the next sentence implies causality, that seems too strong. Generally, it might not be clear to all readers what RSA is and what the link between RSA and encoding models is. Also in the Results section, it would help to motivate or repeat the question/problem that is being addressed in this section.)

AUTHOR RESPONSE:

We thank the Reviewer for this comment. As suggested by the Reviewer, we have supplemented the details in the revised manuscript (page 2, line 12-16):

“Several studies^{18,19} have utilized representational similarity analysis²⁰ to compare the dissimilarity patterns of CNN and fMRI representations, revealing that the human visual cortex shares similar hierarchical representations to CNNs. As a result, CNN-based encoding models have become widely used and have demonstrated excellent performance^{10,12-14}.”

Moreover, we repeated the problem to be addressed at the beginning of each section of the Results:

“To predict the fMRI responses evoked by handwritten characters, ...”; “To verify the encoding performance of the proposed approach on handwritten digit stimuli, ...”; “To assess the feasibility of the proposed approach for encoding natural image stimuli, ...”; “The image reconstruction task aims to reconstruct the images perceived by the participant from the fMRI responses. Based on the pre-trained encoding models, we accomplished this task on handwritten character, handwritten digit, and colorful natural image datasets”; “The image identification task aims to identify the image seen by the participant from the fMRI responses, and this task was accomplished on the grayscale natural image dataset”.

REVIEWER COMMENTS:

(6) The figures need to be improved. There is no axis label or color bar of the intensity plot in Figure 2b. Further, the x-label in Figure 5b is missing. And in Figure 5a, instead of ‘correlation (r)’, it would be helpful to use ‘Subject 1’ and ‘Subject 2’ as subtitle and move ‘correlation’ vertically next to color bar. Moreover, the description of error bars is missing in the legends of the plots.

AUTHOR RESPONSE:

We thank the Reviewer for these constructive suggestions. We have modified the figures based on the Reviewer’s suggestions. The description of error bars is added in the legends of Fig. 2: “The bar charts represent the mean \pm SEM (standard error of the mean) of the encoding accuracies”.

Fig. 2b. The mean stimulus intensities in the train set of the handwritten character dataset.

Fig. 4. Image identification results on natural image dataset. The correlation maps of the measured and predicted fMRI responses to test set images for the two participants. The element at the m_{th} column and n_{th} row represents the correlation between the measured fMRI response for the m_{th} image and the predicted fMRI response for the n_{th} image.

Fig. 5b. The identification accuracies with different numbers of voxels for the two subjects in grayscale natural image dataset.

REVIEWER COMMENTS:

(7) There are many typos / grammatical errors in the manuscript that need to be fixed.

AUTHOR RESPONSE:

We feel sorry for our carelessness. We have checked the whole manuscript carefully and corrected the typos / grammatical errors in our revised manuscript.

REVIEWER COMMENTS:

(8) Abbreviation PCCs in line 88 has not yet been introduced.

AUTHOR RESPONSE:

We are sorry for this careless mistake. We have fixed this issue in the revised manuscript (page 3, line 20-22): “The encoding performance was measured using Pearson's correlation coefficients (PCC) between the predicted and measured responses to the test set images”.

Reviewer 3**REVIEWER COMMENTS:**

The paper addresses neural encoding (prediction of brain response to external stimuli) using unsupervised spiking convolutional neural networks (SCNNs).

The results shown in the paper are relevant for the field, and demonstrate the suitability of the SCNNs for all the tasks analyzed in the paper, which include encoding in symbolic (handwritten characters) and natural images, image reconstruction and identification.

The statistical analysis seems appropriate.

AUTHOR RESPONSE:

We thank the Reviewer for the positive comments regarding our paper. We have now revised the paper substantially and we believe these updates substantially improve the manuscript.

REVIEWER COMMENTS:

(1) One major concern remains regarding reproducibility. The involved methods involve the use of STDP in convolutional neurons. This algorithm, as any (especially unsupervised) ML approach, may highly depend on initial values. Also, STDP has several parameters (which modulate the long-term potentiation/depression). Also, the SCNN shown contains an important first layer constituted by Difference of Gaussians responses. The authors do not disclose their choices for the parameters of these methods, which are key for research reproduction. The level of detail provided is insufficient for reproducibility.

AUTHOR RESPONSE:

We thank the Reviewer for this valuable comment. We agree that the reproducibility is very important for neuroscience. We have made several improvements to address this issue. Firstly, we validated our model on another two datasets^{1,2} in our revised manuscript. Secondly, we supplemented the reproducibility analysis to investigate the effect of initial values on the grayscale natural image dataset (page 9, line 7-15):

“**Reproducibility analysis.** In the proposed encoding model, the unsupervised SCNN was utilized to extract features of the visual stimuli, and the training process of SCNN

was influenced by its initial values. To investigate the impact of initial values on the encoding performance, we trained another SCNN with different initial values on the grayscale natural image dataset and compared its encoding performance with the original one. For each subject, the top 500 voxels with the highest encoding performance were selected for comparison, and no significant differences were observed between the two encoding results (subject1: $p=0.1$, subject2: $p=0.47$, two-tailed two sample t-test)".

Lastly, the hyper-parameters of the architecture of SCNN and STDP were selected based on previous studies^{21,22}, and we have added the explanation for this in the revised manuscript (page 11, line 16-18): "Third, the parameters of STDP and network architecture were selected from previous works^{21,22}, and the impact of different parameters on encoding performance requires further exploration".

We thank the Reviewers and Editors again for their helpful comments and suggestions. We feel that these revisions have very significantly strengthened the manuscript, and hope that it will now be found acceptable for publication in the *Communications Biology*. Thank you very much for your kind consideration.

Sincerely yours,

Huafu Chen, Prof, PhD School of Life Science and Technology
University of Electronic Science and Technology China
Chengdu 610054, P.R. China
Tel: 013808003171
E-mail: chenhf@uestc.edu.cn

Rong Li, PhD School of Life Science and Technology
University of Electronic Science and Technology China
Chengdu 610054, P.R. China
Tel: 013678078624
E-mail: rongli1120@gmail.com

Hongmei Yan, Prof, PhD School of Life Science and Technology
University of Electronic Science and Technology China
Chengdu 610054, P.R. China
Tel: 018628051813
E-mail: hmyan@uestc.edu.cn

References

- 1 Van Gerven, M. A., De Lange, F. P. & Heskes, T. Neural decoding with hierarchical generative models. *Neural computation* **22**, 3127-3142 (2010).
- 2 Horikawa, T. & Kamitani, Y. Generic decoding of seen and imagined objects using hierarchical visual features. *Nature Communications* **8**, 15037, doi:10.1038/ncomms15037 (2017).

- 3 Nishimoto, S. *et al.* Reconstructing visual experiences from brain activity evoked by natural movies. *Current biology : CB* **21**, 1641-1646, doi:10.1016/j.cub.2011.08.031 (2011).
- 4 Miyawaki, Y. *et al.* Visual Image Reconstruction from Human Brain Activity using a Combination of Multiscale Local Image Decoders. *Neuron* **60**, 915-929, doi:<https://doi.org/10.1016/j.neuron.2008.11.004> (2008).
- 5 Du, C., Du, C., Huang, L. & He, H. Reconstructing Perceived Images From Human Brain Activities With Bayesian Deep Multiview Learning. *IEEE Transactions on Neural Networks and Learning Systems* **30**, 2310-2323, doi:10.1109/TNNLS.2018.2882456 (2019).
- 6 Seeliger, K., Güçlü, U., Ambrogioni, L., Güçlütürk, Y. & van Gerven, M. A. J. Generative adversarial networks for reconstructing natural images from brain activity. *NeuroImage* **181**, 775-785, doi:<https://doi.org/10.1016/j.neuroimage.2018.07.043> (2018).
- 7 Wang, W., Arora, R., Livescu, K. & Bilmes, J. in *International conference on machine learning*. 1083-1092 (PMLR).
- 8 Han, K. *et al.* Variational autoencoder: An unsupervised model for encoding and decoding fMRI activity in visual cortex. *NeuroImage* **198**, 125-136, doi:<https://doi.org/10.1016/j.neuroimage.2019.05.039> (2019).
- 9 Kay, K. N., Winawer, J., Mezer, A. & Wandell, B. A. Compressive spatial summation in human visual cortex. *Journal of neurophysiology* **110**, 481-494, doi:10.1152/jn.00105.2013 (2013).
- 10 Allen, E. J. *et al.* A massive 7T fMRI dataset to bridge cognitive neuroscience and artificial intelligence. *Nature Neuroscience* **25**, 116-126, doi:10.1038/s41593-021-00962-x (2022).
- 11 Lage-Castellanos, A., Valente, G., Formisano, E. & De Martino, F. Methods for computing the maximum performance of computational models of fMRI responses. *PLoS computational biology* **15**, e1006397, doi:10.1371/journal.pcbi.1006397 (2019).
- 12 Güçlü, U. & van Gerven, M. A. Deep Neural Networks Reveal a Gradient in the Complexity of Neural Representations across the Ventral Stream. *The Journal of neuroscience : the official journal of the Society for Neuroscience* **35**, 10005-10014, doi:10.1523/jneurosci.5023-14.2015 (2015).
- 13 Wen, H. *et al.* Neural Encoding and Decoding with Deep Learning for Dynamic Natural Vision. *Cerebral cortex (New York, N.Y. : 1991)* **28**, 4136-4160, doi:10.1093/cercor/bhx268 (2018).
- 14 Khosla, M., Ngo, G. H., Jamison, K., Kuceyeski, A. & Sabuncu, M. R. Cortical response to naturalistic stimuli is largely predictable with deep neural networks. *Science Advances* **7**, eabe7547, doi:doi:10.1126/sciadv.abe7547 (2021).
- 15 Izhikevich, E. M. Simple model of spiking neurons. *IEEE Transactions on Neural Networks* **14**, 1569-1572, doi:10.1109/TNN.2003.820440 (2003).
- 16 Victor, J. D., Purpura, K., Katz, E. & Mao, B. Population encoding of spatial frequency, orientation, and color in macaque V1. *Journal of neurophysiology* **72**, 2151-2166, doi:10.1152/jn.1994.72.5.2151 (1994).

- 17 Dumoulin, S. O. & Wandell, B. A. Population receptive field estimates in human visual cortex. *NeuroImage* **39**, 647-660, doi:<https://doi.org/10.1016/j.neuroimage.2007.09.034> (2008).
- 18 Khaligh-Razavi, S. M. & Kriegeskorte, N. Deep supervised, but not unsupervised, models may explain IT cortical representation. *PLoS computational biology* **10**, e1003915, doi:10.1371/journal.pcbi.1003915 (2014).
- 19 Cichy, R. M., Khosla, A., Pantazis, D., Torralba, A. & Oliva, A. Comparison of deep neural networks to spatio-temporal cortical dynamics of human visual object recognition reveals hierarchical correspondence. *Scientific reports* **6**, 27755, doi:10.1038/srep27755 (2016).
- 20 Kriegeskorte, N. & Kievit, R. A. Representational geometry: integrating cognition, computation, and the brain. *Trends Cogn Sci* **17**, 401-412, doi:10.1016/j.tics.2013.06.007 (2013).
- 21 Mozafari, M., Ganjtabesh, M., Nowzari-Dalini, A., Thorpe, S. J. & Masquelier, T. Bio-inspired digit recognition using reward-modulated spike-timing-dependent plasticity in deep convolutional networks. *Pattern Recognition* **94**, 87-95, doi:<https://doi.org/10.1016/j.patcog.2019.05.015> (2019).
- 22 Kheradpisheh, S. R., Ganjtabesh, M., Thorpe, S. J. & Masquelier, T. STDP-based spiking deep convolutional neural networks for object recognition. *Neural networks : the official journal of the International Neural Network Society* **99**, 56-67, doi:10.1016/j.neunet.2017.12.005 (2018).

REVIEWERS' COMMENTS:

Reviewer #1 (Remarks to the Author):

The authors have satisfactorily addressed the issues raised from the previous round, and made considerable improvements to the manuscript. I have no further comments.

Reviewer #2 (Remarks to the Author):

The authors have meticulously addressed all my comments and have carefully revised the manuscript. I especially appreciate their effort to compute the noise ceiling for one of the fMRI datasets. I believe the current version of the manuscript has been improved significantly and I have no further suggestions at this point. I commend the authors for their work.

Reviewer #3 (Remarks to the Author):

The authors have improved the quality of the paper by addressing the issues pointed out during the revision of the paper. The paper overall contains a good comparison of unsupervised models based on spiking neurons (Integrate and Fire in this study), with linear regression on top (supervised) to handle encoding and classification tasks.

Although the results are very promising for the application of SNNs to these tasks, the comparisons should be framed carefully: the structure of the non-spiking neural networks used as baseline are constrained, as the authors mention, to be consistent with the structure of the SNNs. So, the comparisons are not made with the best possible non-spiking counterpart. The limitations in the SNN structure are probably derived from the nature of the STDP training procedures, and that is why the constraints from SNNs are applied to non-SNNs, and not the other way. I also miss the application of discriminative training techniques to SNNs, which would make possible more complex structures for the SNNs, while losing biological plausibility.

However, the work is a good study of the application of unsupervised techniques for low level encoding.

University of Electronic Science and
Technology of China

Huafu Chen, PhD

Director, MOE Key Lab for Neuroinformation
The Clinical Hospital of Chengdu Brain Science Institute
School of Life Science and Technology
University of Electronic Science and Technology of China
Chengdu 610054, P.R. China
Tel: 013808003171
E-mail: chenhf@uestc.edu.cn

RE: COMMSBIO-23-0421 A

August 6, 2023

Reviewer 1

REVIEWER COMMENTS:

The authors have satisfactorily addressed the issues raised from the previous round, and made considerable improvements to the manuscript. I have no further comments.

AUTHOR RESPONSE:

We would like to thank the Reviewer for the positive feedback. We greatly appreciate your time and effort in reviewing our manuscript. We are glad to hear that our revisions have addressed the previous concerns to your satisfaction.

Reviewer 2

REVIEWER COMMENTS:

The authors have meticulously addressed all my comments and have carefully revised the manuscript. I especially appreciate their effort to compute the noise ceiling for one of the fMRI datasets. I believe the current version of the manuscript has been improved significantly and I have no further suggestions at this point. I commend the authors for their work.

AUTHOR RESPONSE:

Thanks for your support in advancing scientific knowledge through the peer-review process. We look forward to seeing our work contribute to the scientific community.

Reviewer 3

REVIEWER COMMENTS:

The authors have improved the quality of the paper by addressing the issues pointed out during the revision of the paper. The paper overall contains a good comparison of unsupervised models based on spiking neurons (Integrate and Fire in this study), with linear regression on top (supervised) to handle encoding and classification tasks.

Although the results are very promising for the application of SNNs to these tasks, the comparisons should be framed carefully: the structure of the non-spiking neural networks used as baseline are constrained, as the authors mention, to be consistent with the structure of the SNNs. So, the comparisons are not made with the best possible non-spiking counterpart. The limitations in the SNN structure are probably derived from the nature of the STDP training procedures, and that is why the constraints from SNNs are applied to non-SNNs, and not the other way. I also miss the application of discriminative training techniques to SNNs, which would make possible more complex structures for the SNNs, while losing biological plausibility.

However, the work is a good study of the application of unsupervised techniques for low level encoding.

AUTHOR RESPONSE:

We thank the Reviewer for the positive feedback. As mentioned by the Reviewer, the comparisons of SCNNs and CNNs were limited to shallow structures, which were derived from the nature of the STDP algorithm. We have also addressed this issue in the Discussion section (page 7, line 27-33):

“Despite the progress made in neural encoding using SCNN, there remain several limitations. First, the architectures of SNNs are typically shallower than those of deep-learning networks, which restricts their ability to extract complex and hierarchical visual features. Recent studies have attempted to address this issue and have made some headway¹⁻³. The incorporation of a deeper SCNN into our model would further enhance encoding performance and enable investigation of the hierarchical structure of the visual cortex”.

Exploring various training strategies for SCNN-based encoding models represents a crucial direction for our future research, and, like the Reviewer, we are eager to uncover novel insights and discoveries in this area.

We thank the Reviewers and Editors again for their helpful comments and suggestions. We feel that these revisions have very significantly strengthened the manuscript, and hope that it will now be found acceptable for publication in the *Communications Biology*. Thank you very much for your kind consideration.

Sincerely yours,

Huafu Chen, Prof, PhD School of Life Science and Technology
University of Electronic Science and Technology China
Chengdu 610054, P.R. China
Tel: 013808003171
E-mail: chenhf@uestc.edu.cn

Rong Li, PhD School of Life Science and Technology
University of Electronic Science and Technology China
Chengdu 610054, P.R. China
Tel: 013678078624
E-mail: rongli1120@gmail.com

Hongmei Yan, Prof, PhD School of Life Science and Technology
University of Electronic Science and Technology China
Chengdu 610054, P.R. China
Tel: 018628051813
E-mail: hmyan@uestc.edu.cn

References

- 1 Mozafari, M., Ganjtabesh, M., Nowzari-Dalini, A., Thorpe, S. J. & Masquelier, T. Bio-inspired digit recognition using reward-modulated spike-timing-dependent plasticity in deep convolutional networks. *Pattern Recognition* **94**, 87-95, doi:<https://doi.org/10.1016/j.patcog.2019.05.015> (2019).
- 2 Kheradpisheh, S. R., Ganjtabesh, M., Thorpe, S. J. & Masquelier, T. STDP-based spiking deep convolutional neural networks for object recognition. *Neural networks : the official journal of the International Neural Network Society* **99**, 56-67, doi:10.1016/j.neunet.2017.12.005 (2018).
- 3 Wu, Y., Deng, L., Li, G., Zhu, J. & Shi, L. Spatio-Temporal Backpropagation for Training High-Performance Spiking Neural Networks. *Front Neurosci* **12**, 331, doi:10.3389/fnins.2018.00331 (2018).